# A Guide to Applying the Sex-Gender Perspective to Nutritional Genomics

**DOI:** 10.3390/nu11010004

**Published:** 2018-12-20

**Authors:** Dolores Corella, Oscar Coltell, Olga Portolés, Mercedes Sotos-Prieto, Rebeca Fernández-Carrión, Judith B. Ramirez-Sabio, Vicente Zanón-Moreno, Josiemer Mattei, José V. Sorlí, Jose M. Ordovas

**Affiliations:** 1CIBER Fisiopatología de la Obesidad y Nutrición, Instituto de Salud Carlos III, 28029 Madrid, Spain; oscar.coltell@uji.es (O.C.); Olga.Portoles@uv.es (O.P.); Rebeca.Fernandez@uv.es (R.F.-C.); jose.sorli@uv.es (J.V.S.); 2Department of Preventive Medicine and Public Health, School of Medicine, University of Valencia, 46010 Valencia, Spain; 3Department of Computer Languages and Systems, Universitat Jaume I, 12071 Castellón, Spain; 4School of Applied Health Sciences and Wellness, Ohio University, Athens, OH 45701, USA; sotospri@ohio.edu; 5Department of Nutrition, Harvard T.H. Chan School of Public Health, Boston, MA 02115, USA; jmattei@hsph.harvard.edu; 6Department of Environmental Health, Harvard T.H. Chan School of Public Health, Boston, MA 02115, USA; 7Oncology Department, Sagunto Hospital, 46500 Sagunto, Spain; jbramire@uv.es; 8Ophthalmology Research Unit “Santiago Grisolia”, Dr. Peset University Hospital, 46017 Valencia, Spain; Vicente.Zanon-Moreno@uv.es; 9Red Temática de Investigación Cooperativa OftaRed, Instituto de Salud Carlos III, 28029 Madrid, Spain; 10Nutrition and Genomics Laboratory, JM-USDA Human Nutrition Research Center on Aging at Tufts University, Boston, MA 02111 USA; jose.ordovas@tufts.edu; 11Department of Cardiovascular Epidemiology and Population Genetics, Centro Nacional de Investigaciones Cardiovasculares (CNIC), 28029 Madrid, Spain; 12IMDEA Alimentación, 28049 Madrid, Spain

**Keywords:** sex, gender, diet, nutritional genomics, nutrigenomics, precision nutrition

## Abstract

Precision nutrition aims to make dietary recommendations of a more personalized nature possible, to optimize the prevention or delay of a disease and to improve health. Therefore, the characteristics (including sex) of an individual have to be taken into account as well as a series of omics markers. The results of nutritional genomics studies are crucial to generate the evidence needed so that precision nutrition can be applied. Although sex is one of the fundamental variables for making recommendations, at present, the nutritional genomics studies undertaken have not analyzed, systematically and with a gender perspective, the heterogeneity/homogeneity in gene-diet interactions on the different phenotypes studied, thus there is little information available on this issue and needs to be improved. Here we argue for the need to incorporate the gender perspective in nutritional genomics studies, present the general context, analyze the differences between sex and gender, as well as the limitations to measuring them and to detecting specific sex-gene or sex-phenotype associations, both at the specific gene level or in genome-wide-association studies. We analyzed the main sex-specific gene-diet interactions published to date and their main limitations and present guidelines with recommendations to be followed when undertaking new nutritional genomics studies incorporating the gender perspective.

## 1. Importance of Sex/Gender Differences in Precision Medicine and Precision Nutrition

In recent years, the concept of precision medicine or what is also called personalized medicine [1] has become increasingly important. With the publication by Francis Collins [2], director of the United States National Institutes of Health (NIH), announcing the new era of Precision Medicine and the setting up of a nationwide cohort aimed at recruiting and undertaking follow-up on a million individuals in order to generate omic and other types of data for precision medicine, research with this precision perspective has intensified in all disciplines internationally. The field of nutrition has also felt the general influence of the aims of precision medicine and, in recent years, has strongly developed the concept of precision nutrition or personalized nutrition.

### 1.1. Precision Nutrition and Sex/Gender Differences

Currently, there is no single definition of precision nutrition, and although in most cases precision nutrition is used as a synonym for personalized nutrition, several authors claim that there may be a number of differences in the sense that precision nutrition could go further afield than personalized nutrition [3]. In general, we can define precision nutrition as that which uses information on the characteristics of an individual, including omic markers, to make more personalized recommendations that could contribute to preventing or treating a disease in the best way possible [3,4,5]. Therefore, sex/gender, in addition to the other markers, is one of the most relevant variables for precision nutrition.

For precision nutrition, it is essential to have omic markers available and not only genomic ones, but increasingly epigenomic, metabolomics, transcriptomic and metagenomic ones, among others [5]. Furthermore, precision nutrition needs to have prior evidence gathered by nutrigenetic and nutrigenomic studies available in order to make specific recommendations (Figure 1).

### 1.2. Nutritional Genomics and Sex/Gender Differences

Precision nutrition cannot be undertaken without having prior findings available from nutrigenetic and nutrigenomic research. Both disciplines form part of nutritional genomics and are often used synonymously, but a difference can be established between them. Whereas nutrigenetics studies the different response to diet depending on the genotype of the individual, nutrigenomics studies the mechanisms through which different responses to diet, depending on genotype take place, integrating various omics [6]. Despite the hundreds of studies published on nutritional genomics [7] in recent years, much more research is required to attain a sufficient level of evidence for the knowledge required to allow its practical application to precision nutrition. In general, the studies conducted so far have presented low-level consistency when analyzing similar gene-diet interactions. Various factors may have contributed to that, among them, and quite apart from problems related to study design, are those associated with the characteristics of the population. Even though nutrition genomics seeks to generate more specific results to make more personalized nutritional recommendations, these studies have focused on genotype and other omic markers and have not paid sufficient attention to other critical factors for personalizing diets such as age and sex/gender, among others. Having not paid attention to the sex/gender variable of the participants may have contributed to different studies analyzing the same gene-diet interactions and producing discordant results, especially if each of those studies had included a different percentage of men and women and there had been a certain heterogeneity per sex. Hence, the need to know the characteristics of the individuals better in with precision nutrition, it is essential to conduct new nutritional genomic studies that take into account the sex/gender of the participants. Thus, in this guide, we will present the current situation analysis from the gender perspective in nutritional genomic studies, primarily in nutrigenetics, but also mentioning several nutrigenomic aspects, focusing on the different intermediate and final phenotypes of the leading chronic diseases related to nutrition. To do so, we will define the sex/gender perspective and its specific application to nutritional genomic studies and present examples of studies that have tackled the sex/gender perspective with greater or lesser complexity, as well as examples of gene-sex-diet interactions and review their advantages and limitations. We will also analyze the advances made in identifying biological sex, discuss the difficulty of genomic studies that analyze genes in the sexual chromosomes by presenting several examples. Finally, we will make a series of specific recommendations to be taken into account in the form of a guide to analyze data from previously undertaken nutritional genomic studies and fundamentally for designing and conducting new nutritional genomic studies with a sex/gender perspective.

## 2. General Context of Differences Per Sex and Gender Perspective in Biomedical Research and Nutritional Genomics

Historically, women have had fewer opportunities in society than men [8]. The scientific field has not been exempt from that, not only because most researchers and leaders have been men, but also because, when it comes to biomedical research in humans, mainly through clinical trials, fewer women have been analyzed than men. This means that many conclusions of biomedical studies have been fundamentally obtained using results of what happens in men, and the results have been generalized to directly include women without specifically analyzing them and without knowing whether their response is different from men or not [9]. Thus, it has only been in recent decades, faced with the claim that important differences may exist in risk factors, symptomology, treatment and prognosis of various diseases between men and women, that both sexes have been studied in greater detail [10,11,12,13,14,15]. As important differences in various factors related to health and disease between men and women are increasingly noted, specialist researchers and health planners are insisting on the increasing need to carry out studies that include and compare men and women in order to better understand similarities and differences and to be able to apply that knowledge effectively to disease prevention and treatment [16]. It is becoming increasingly evident that ignoring the influence of sex in research compromises the validity and generalizability of the findings and contributes to bringing about a stagnation or even regression instead of making progress on the level of evidence and scientific knowledge. Hence, in recent years there has been an attempt to promote epidemiological studies that adopt the sex/gender perspective, analyzing at least differences between men and women separately when the study includes both [17,18].

Presenting results stratified by men and women allows us to have information available for conducting meta-analyses and showing, with a higher level of evidence, whether or not differences in the factor and/or disease studied exist between men and women. Nevertheless, few studies exist that have presented results stratified by sex and, therefore, it is still complicated to conduct meta-analyses of the studies published to obtain conclusions on this issue. As an example of this situation, the conclusions arrived at in a meta-analysis carried out a few years ago by Doull et al. [19] to examine the use of sex/gender analyses in systematic reviews of cardiovascular health. They chose a random sample of 38 Cochrane systematic reviews addressing interventions for cardiovascular diseases from 2001 to 2007. The general conclusion was that a sex/gender analysis was not considered in the reviews examined and only 2 of 38 reviews reported any sex/gender research gaps. Moreover, where sex or gender was mentioned, the terms were used interchangeably. The authors, therefore, insisted on the need to promote the undertaking of original studies of men and women in order to make data available at this level and, in parallel, to also promote the undertaking of systematic reviews and meta-analyses using the sex/gender perspective. Without this information, it would not be possible to build a robust evidence base for future analysis of evidence for decision-making.

One way of encouraging studies to be undertaken on men and women (or males and females in studies on animals or cells) and to present separate data is for study-funding agencies to prioritize that undertaking. In this context, The National Institutes of Health (NIH) Office of Research in Women’s Health in the United States, under the efforts of Janine Clayton has issued a mandate to close the knowledge gap on women’s health and sex/gender research in epidemiological studies in humans as well as in the basic research arena [20,21]. These efforts, begun in 2014 and 2015, are having a substantial impact on all related fields, from establishing guidelines on conducting and evaluating studies [22,23,24]. In these recommendations, particular emphasis has been placed on basic research studies and animal models, given that traditionally these studies have not reported on the sex of animals, cells, or tissues used, so contributing to the ambiguity of knowledge that now exists on the influence of sex in these studies. The announcement of the NIH in May 2014 insisted that studies not only had to differentiate per sex, but to go further and specifically ensure that the investigators account for sex as a basic biological variable (SABV) in NIH-funded preclinical research [23]. In Europe, concern about the inclusion of a gender perspective in financed research began somewhat earlier. Quite apart from the pioneering initiatives in several countries, such as Sweden and Switzerland, we should also point out that the gender perspective has been a major initiative of the European Union gender equality policy for research.

Gender perspective has many definitions, among which we should mention that made by the Swiss Center of Expertise in Human Rights in 1997 and that has been taken up by the European Institute of Gender Equality (EIGE) that defines it as a “Perspective taking into account gender-based differences when looking at any social phenomenon, policy or process”. This perspective was applied to research for the first time under Framework Program 5 (FP5). In 2000–2001, the European Commission (EC) commissioned a series of Gender Impact Assessment (GIA) studies on FP5 to assess, among other aspects, whether the lines of research prioritized in FP5 affected men and women differently [25]. This action aimed to obtain conclusions to be included in a gender perspective for the following program (FP6). One of these GIA studies focused on the “Quality of Life and Management of Living Resources” and had a subsequently wide impact given that its recommendations on the importance of gender in research were later introduced for FP6 applicants [25]. Within this program, the so-called GenderBasic Project (2005–2008) was conducted, whose main aim was to provide scientists involved in health related research (including both basic and clinical research) funded by the EU, with practical tools and best practices regarding sex/gender analysis in those projects. The outcome of this project was the publication of 10 reviews in the journal Gender Medicine entitled “Bringing Gender Expertise to Biomedical and Health-Related Research” [26], as well as making recommendations on promoting the gender perspective in the following PF7 [25]. One of the reviews published in the Gender Medicine special was conducted by Prof. José Ordovás, and focused on the influence of gender in gene-environment interactions [27]. It was later, in 2013, when the analysis of sex/gender variables in the research projects of the EU [28] was prioritized still further. Among the more pioneering countries in encouraging sex/gender analyses in research projects in Canada, which in 2010, had already established several formal recommendations for research projects [29]. In parallel, different public and private bodies in various countries have been incorporating the gender perspective into their calls for projects by positively evaluating this approach in their criteria, both in basic and clinical research. All these initiatives directly related to introducing the gender perspective into biomedical research would not have been possible without the immense prior work of pioneering individuals, institutions and bodies on the gender perspective in society as a whole, and are too numerous to mention here in greater detail. On the international level we can mention the policy on gender perspective of the World Health Organization (WHO), as in the document “Integrating gender perspectives in the work of WHO: WHO gender policy”, published in the year 2002 [30]. Table 1 provides various Web links containing information related to gender perspective in the most important institutions and bodies of the same.

In addition to these considerations, another aspect to be taken into account is that in clinical trials people decide on whether or not they want to participate in these studies. In an ideal world, it would be great to have a mix of men and women. However, if the research focuses on breast cancer, for example, which affects many more women than men, it would be practically impossible to achieve a similar number of both men and women participants. Nonetheless, despite these limitations, and apart from when health problems mostly affect only one of the sexes, efforts have to be made to optimize the representation of men and women in studies.

Currently the concept of Gender Mainstreaming is being implemented as an international strategy to ensure gender equality [31]. According to EIGE, Gender Mainstreaming: 

“involves the integration of a gender perspective into the preparation, design, implementation, monitoring and evaluation of policies, regulatory measures and spending programs, intending to promoting equality between women and men, and combating discrimination” [31].

## 3. Difference Between Sex and Gender

So far in this work, we have used the words sex and gender without pointing out the difference, but we are now going to make the distinction. Among the less expert research into gender perspective, there is confusion in the meanings of both words. Most scientific works have used sex and gender as synonyms which has added to the confusion. Although there is no total agreement between different authors, there is more of an agreement on the definition of sex than on gender. Table 2 shows various definitions of sex and gender with their respective references [32,33,34,35,36]. In general, sex is defined as a biological construct based on two different chromosomal configurations (XX as opposed to XY) and on a set of biological processes related to sexual beings. In analyses designed to study differences per sex, this is treated as a dichotomous variable. However, gender is a much more complicated variable and is defined as a socio-cultural and political construction that determines the relationship between people, granting benefits and access to resources to those that find themselves in higher positions in a gender hierarchy.

Thus, gender would not be a dichotomous variable but a continuous variable because it defines psychological, behavioral, cultural and political characteristics that are expressed in a wide range of situations, which, moreover, can be dynamic. Both men and women can present a range of different gender points and can even be overlapping. According to these gender definitions, a person’s sex and gender may not necessarily be the same. Some people can identify themselves as being of a different gender to the sex they were born with. There can even be people who, during one stage of their lives identified themselves more with the masculine gender and in another with the feminine gender [36]. Thus, given these definitions, it is not correct to use sex and gender as synonyms in health research.

Various guides have been published aimed at directing researchers, editors and the reviewers of different journals on how to analyze data, present results and make conclusions to prepare articles for publications in journals. Among these are the SAGER (Sex and Gender Equity in Research) Guidelines [37] that contain very detailed information on the different aspects to integrate sex and gender assessment into manuscripts.

### 3.1. Sex and Gender Identification in Studies on Nutritional Genomics

To apply the gender perspective to nutritional genomics studies we need to obtain information on the sex and gender variables. A priori, sex has a clearer definition and is easier to measure. However, gender has a more complicated definition, and we cannot always measure it well in nutritional genomics studies due to the lack of information on the other variables required for its definition.

#### 3.1.1. Measuring Sex in Nutritional Genomics Studies

Generally speaking, in nutritional genomics studies, sex is a self-reported variable. Often the man or woman question is included in the questionnaires and the individual identifies himself or herself with one of the two categories. Sometimes it is the researcher who assigns the participant to the male or female sex by their name, appearance, etc. Nevertheless, various definitions of sex such as those presented in Table 2, indicate that sex “refers to biological differences between females and males, including chromosomes, sex organs, and endogenous hormonal profiles” [34]. With regard to differences in chromosomes, a recent work published in JAMA by Janine A Clayton and Cara Tannembaum [38], from the Office of Research on Women’s Health of NIH, Bethesda, recommends that, in an article, the “sex” variable of the participants be included in an initial table on the characteristics of the sample studied, stating the number of male and female participants and using, for that purpose, a genotyping technique on a blood sample of the participants. Although employing a genotyping technique is a good approach to determining sex, and allows us to indirectly assess the karyotype (46,XY (male) or 46,XX (female)), this is not feasible for all studies. Clayton and Tannembaum [38], after recommending the presentation of the “sex” variable as males and females determined by genotyping in the laboratory, also recommend presenting the “gender” variable in the first table, as the number of men and women ascertained by self-report. This recommendation may confuse, as in the normal situations of epidemiological and nutritional genomics studies, sex has been self-reported and not determined genetically.

In terms of genetic sex, various genotyping techniques that determine the sex of an individual from a blood sample have been validated. Initially the amelogenin test (frequently used in forensics) was used, based on determining length polymorphisms in AMEL genes [39]. Our research group used this test to analyze the concordance between the sex determined by genotyping and self-reported sex [39]. We genotyped 1,224 individuals participating in a biomedical study, and the overall concordance rate was 99.84% (1222/1224). Two samples showed a female amelogenin test outcome, being codified as males in the database, and we then identified them as coding errors or sample handling mistakes [39]. Some years ago, without taking the gender perspective into account, we concluded that the undertaking of sex determining analyses by genotyping could be very useful as a means of quality control. However, there is also the possibility that several discrepancies, quite apart from the errors of the amelogenin test itself, are due to individuals with a certain chromosomal sex (for example: XY), reporting a different gender (woman, in this case), but this would not constitute an error at the genotyping level (being a real difference between the biological sex and the reported gender). There are now more advanced techniques for determining sex through next generation sequencing (NGS) [40], or through panels of selected polymorphisms that could easily be used in nutritional genomics studies that have genome-wide genotyping available for quantifying the discrepancies. However, whilst this genotyping possibility is technically available, most studies into nutritional genomics have determined sex through a self-reporting method, instead of by genotyping.

On the suggestion of classifying sex through genotyping techniques, other authors argue that we have to consider whether “sex” should be restricted to reproductive differences between males and females, or extended to include all biological differences between males and females [41,42]. Moreover, researchers in a qualitative study [42] thought that the traditional definition of “sex” was not sufficient for understanding the biological body, stressing “it was difficult to define where ‘sex’ and ‘gender’ started and ended as the concepts were seen as intertwined”.

Finally, with regard to biological sex being determined through different techniques in the laboratory, we have to bear in mind that, although affecting a very small number of people, there are other chromosomal alterations that give rise to more or less undetermined sexes, such as 45,X; 47,XXY; 45,X/46,XY, among others that have been widely discussed in a consensus review [43].

#### 3.1.2. Gender Measurement in Nutritional Genomics Studies

Measuring gender is still more difficult than measuring sex in biomedical research studies and nutritional genomics in particular [16,32,33,34,35,36,41,42,44]. Despite there being many definitions, they generally agree on stating that gender “is a multidimensional social construct that is culturally based and historically specific, and thus constantly changing” [44]. It, therefore, refers to the socially prescribed and experienced dimensions of "femaleness" or "maleness" in a particular society, and is manifested at many levels. Given the difficulty of definition, from a practical point of view it is necessary to know which variables can provide the best information in order to classify individuals into the different genders, either generally or specifically for each type of study. Although several advances have been made [45,46], there is still little consensus on how to measure the factors that determine gender and how many categories of gender can be validly analyzed and compared. As the gender perspective is increasingly incorporated into nutrition genomics studies, better indicators and metrics will be developed to measure the gender variable more accurately. One example of these developments is the work of Pelletier et al. [47] who created a gender-score in order to measure the gender variable in women and men participants in GENESIS-PRAXY (GENdEr and Sex determInantS of cardiovascular disease: From bench to beyond-Premature Acute Coronary SYndrome). First, they selected variables to form part of the scale and with them created a score from 0 to 10 points. A higher score meant a more feminine gender. On using this variable, they observed that patients with characteristics traditionally ascribed to women were more likely to experience a recurrent acute coronary syndrome than patients with characteristics traditionally ascribed to men, independently of biological sex [47]. Similar developments in the field of nutritional genomics are necessary to measure gender better in each type of study.

Other approaches to measuring gender have been based on gender identity and sexual orientation. Thus, a study undertaken on a large sample of college students [48] to discover the relationship between gender and eating-related pathology posed the gender question (female, male and transgender), as well as sexual orientation (heterosexual, gay/lesbian, bisexual or unsure). Using these questions, they created a variable with 7 levels (trans-gender, cis-gender sexual minority men, cis-gender unsure men, cis-gender heterosexual men, cis-gender sexual minority women, cis-gender unsure women, and cis-gender heterosexual women). They observed a higher association between the trans-gender and cis-gender sexual minority and eating disorders.

## 4. Measuring the Difference in Outcomes Depending on Sex/Gender

Nutritional genomics studies the interaction between genes and diet determining the intermediate or final phenotypes of disease. When it comes to designing a study of this type, it is critical to know a priori whether differences exist between men and women in the phenotypes of disease to better direct the genetic and environmental variables to be determined in the sample studied. Although some years ago there was no stratified information on the outcomes, thanks to the influence of research into gender perspective we now have more information revealing these differences. Among them the review of Giovannella Baggio et al. [49] entitled “Gender Medicine: a task of the third millennium”, which discusses the influence of sex/gender in five fields of medicine: Cardiovascular diseases, oncology, liver diseases, osteoporosis and pharmacology. In parallel, other authors have also revealed the enormous influence of gender in ophthalmology [50,51] and on different aspects of the metabolism [52], among other disease phenotypes.

## 5. Gender Perspective in Genotype Analyses in Nutritional Genomics Studies

In nutritional genomics studies a key point is the selection of the genes and gene variants to be included in the analysis. Initially, genes and gene variants, basically single nucleotide polymorphisms (SNPs), were chosen based on candidate genes and a gene variant functionality approach [7]. However, after undertaking genome-wide association studies (GWAS), most nutritional genomics studies have focused on the main SNPs identified through GWAS, either in isolation or jointly by calculating so-called genetic risk scores (GRS) [53]. Nevertheless, most of these GWAS have not taken into account the gender perspective and few studies have undertaken an analysis stratified by men and women. For this reason, we believe that there is still a lack of essential information on the central genes and/or SNPs that specifically can have a differential function in men and women. Having that information would allow us to design better studies on sex-specific gene-diet interactions and to develop so-called sex-specific GRS instead of working with the less specific global GRS, as has been done for other obesity-specific GRS [54].

Besides these considerations, one has to bear in mind when analyzing polymorphisms in the sexual chromosomes that there are limitations in traditional statistical analyses, given that women have two X chromosomes and no Y chromosome. Therefore, the statistical analysis for polymorphisms in chromosomes X and Y are difficult and should be adapted, instead of excluded for the present GWAS. More GWAS including X and Y chromosomes should be promised [55]. Moreover, several X chromosomes in women may escape inactivation, so opening up a new field of research on the epigenetic level [55,56,57]. We shall now move on to review the main findings in detecting SNPs differentially associated with disease phenotypes in men and women, as well as their limitations.

### Sex-specific SNPs Associated with Disease, Current State of the Genome-Wide Association Studies

It is well-known that there are metabolic reactions that are regulated differently in males and females, and therefore, sexual dimorphism is apparent in many diseases such as obesity, diabetes or cardiovascular diseases [58]. However, apart from differences in hormone effects, the XX and XY chromosomes contribute to such variance. For example, using four core genotypes mouse models, Chen et al. [59] found that the sex chromosomes, independently from gonadal sex, play a role in adiposity (more subcutaneous fat deposit in females and visceral in males), feeding behavior, fatty liver, and glucose homeostasis, with higher risk for XX chromosome complement than XY [60]. The contribution of the X chromosome to the haploid human genome is 5% with around 800 protein-encoding genes out of the total 20,000 genes [61]. It seems quite relevant to study then, the influence of sex differences associated with disease. However, while during the genomic era an increasing number of GWAS have tried to detect SNPs associated with diseases, only around 30 % included the X chromosome in their analysis [62]. The challenges associated with the X-chromosome (described in the next section), is probably the reason why few studies are analyzing the association between SNPs in the X chromosome and different traits. In contrast, we found more studies applying sex-stratified analysis in GWAS. Table 3 shows a summary of some markers associated with different traits in chromosome X or markers identified in sex-stratified GWAS [63,64,65,66,67,68,69,70,71,72,73,74,75,76,77,78].

Tukiainen et al. [63] aimed to assess the contribution of 404,862 chromosome X SNPs to cardiometabolic and anthropometric traits in Finnish and Swedish individuals. The authors found that the rs182838724 near *FGF16/ATRX/MAGT1* and the rs1751138 near *ITM2A* were associated with height in females and the rs139163435 in Xq23 with fasting insulin in males. In addition, variants near *ITM2A* showed escape from X chromosome inactivation highlighting the value of accounting for potential sex heterogeneity when assessing chromosome X associations. Another study assessing the male specific region of the human Y chromosome, which has been identified as a candidate for gender-related differences in the development of cardiovascular diseases, found that the TBL1Y(A) USP9Y(A) haplotype of the Y chromosome, present only in people of African origin, was associated with better profiles of lipoprotein patterns [64].

Ohlsson et al. [65] found an SNP in the X chromosome associated with lower concentrations of testosterone. Testosterone levels in males are important because lower concentrations have been associated with cardiovascular mortality, osteoporosis, metabolic syndrome, and type 2 diabetes. These polymorphisms may assist in the identification of males at risk of low serum testosterone, with potential clinical use in future studies [65]. Other SNPs identified include the rs5934507 in the locus Xp22.31 associated with bone metabolism or some X or Y- genes associated with psychiatric disorders [79].

Studies assessing sex-stratified GWAS also provide useful information. Winkler et al. [67] performed a meta-analysis with genome-wide chip and Metabochip from the Genetic Investigation of Anthropometric Traits (GIANT) Consortium to look at sex-specific effects of gene variants on BMI or waist to hip ratio. They found 28 loci with larger effects in females than in males, five in males than in females, and 11 showed opposite effects between sexes and waist to hip ratio [67]. Randall et al. [68] identified three novel anthropometric trait loci (near MAP3K1, HSD17B4, PPARG), all of which had genome-wide significance in females (*p* < 5 × 10(−8)), but not in males. Most of the SNPs identified in GWAS with different effect in males than in females are located in genes that have relevant functions to metabolism related to cardiovascular and associated diseases. For example, the PPARG gene plays a role in diabetes genetics and therapy. Thus, the results from stratified analysis also provide important evidence on the biology underlying sex differences in many important diseases. This is notable since many diseases should be approached differently based on gender differences, and clinically can have a meaningful impact on the treatment of diseases. Although the research is emerging, the sex-gene specific associations are clearly understudied and, given the implications, there is an imperative need to study and treat disease through sex/gender informed research.

#### Methodology/Limitations

Techniques to evaluate and identify SNPs have improved considerably over the last decade. Nowadays, platforms such as Affymetrix Axiom Exome Genotyping Arrays, or the Illumina HumanOmni5-Quad BeadChip include hundreds of thousands or millions of markers and from those, some thousands or less (depending on the company) are specific to chromosome X. Many previous GWAS studies have simply not included sex chromosomes in their analyses, despite being assayed on all GWAS microarray platforms [62]. What was different in the past was that genotyping chips contained very few X-chromosome markers. Still several reasons have been given for why variants in the X chromosome are difficult to assess compared to autosomal chromosomes and why they have not been included in GWAS analyses. These include lack of coverage on GWAS chips, differences in the number of genes or variants on the X chromosome compared to the autosomes. Females have two copies of X chromosome and males only one, thus males might cluster differently from females, and so the statistical analysis and interpretation of the X chromosome can be a challenge. Differences in the minor-allele frequency of variants on the X chromosome (the expected frequencies are sex dependent), or the fact that current standard sequencing technologies cannot distinguish variants in the silenced X chromosome provide other reasons (random X inactivation in women could neglect important associations) [61,62]. Further limitations include the lack of powerful statistical tests and challenges analyzing data due to previous complications. Chen and colleagues [80] proposed a new model of analysis, Xcat, that makes the assumption that the at-risk allele is the same for both males and females if the SNP on the X chromosome is associated with the disease for both genders. However, there is no consensus on how to handle X chromosome information and each study has used different methods. For example, some studies have accounted for Hardy-Weinberg equilibrium, X-linked SNPs, and X inactivation [81,82], but very few of the GWAS have used specific methods other than PLINK (that does not take into account X inactivation) [83]. As previously discussed, those challenges have led to an increased number of sex-stratified analyses in autosome chromosomes rather than X chromosomes.

To summarize, based on the evidence there are still very few studies assessing the variants in the X chromosome. Neglecting the inclusion of variants in the sex-chromosomes can certainly increase disparities and misinformation on how to better apply knowledge for personalized treatments and counseling of a disease, given that sex and gender are critical determinants of health [14,15,16,84].

## 6. Specific Sex/Gender Gene-Diet Interactions

Most studies carried out on nutritional genomics have analyzed gene-diet interactions without taking the gender perspective into account. Various factors have contributed to that, but fundamentally we could point to the lack of awareness among researchers of the importance of study design and the corresponding analyses, taking that perspective into account. Despite this, several published studies have indeed analyzed gene-diet interactions in men and women separately and which have found several specific interactions. Table 4 presents in greater detail some of the more relevant studies that have found sex-specific associations in nutritional genomics [85,86,87,88,89,90,91,92,93,94,95,96,97,98,99,100,101,102,103,104,105]. In general, great heterogeneity is detected when it comes to presenting results, as many studies do not undertake a formal statistical analysis of the sex-gene-diet interaction term, limiting themselves to presenting the results of the stratified analyses on men and women, without taking into account the statistical significance of the interaction term indicating the statistical heterogeneity and justifying or not a stratified analysis. Often the problem of not carrying out a formal analysis of the sex-gene-diet interaction term resides in the sample size, given that it is necessary to analyze a large sample size in order to be able to detect an interaction term as statistically significant on that level, and it is not always possible to achieve the high number of participants required. On other occasions, it is simply that the authors did not think about carrying out that calculation. Therefore, greater standardization is required on how to undertake studies to increase the quality of the results obtained.

Out of the sex-specific associations presented in Table 4, it is difficult to know how many of them are truly physiological differences depending on sex, or how many are simply the result of statistical false positives (type I errors), as on undertaking the analyses of two groups, the probability of finding a statistically significant association purely by chance is increased. This fact, as well as the increased cost of studies by having to include a greater sample size of men and women to achieve greater statistical power, has often been criticized for limiting the incorporation of the gender perspective [25]. However, an improvement in study design and statistical analysis plan can minimize those problems. Moreover, the possibility of introducing other complementary omics (epigenomics, transcriptomics, metabolomics, etc.) and collaborations with other researchers to try to replicate the results obtained will be crucial for validating sex-specific gene-diet interactions and distinguishing them from false negatives in the analyses. Most of the studies presented in Table 4 focus on gene-diet interactions by analyzing gene variants [85,86,87,88,89,90,91,92,93,94,95,96,97,98,99,100], as these are the type of studies that have mostly been published to date. However, with the development of omics sex-specific gene-diet interactions at the epigenomic level are now being analyzed, for example, those studies reporting on the methylation level in Table 4 [101,102]. We have also included in Table 4 other studies that have found specific sex differences in the changes that diet produces in the microbiome/metagenome [103,104]. All these studies will increase in the coming years, and it is essential that they incorporate the gender perspective given these initial differences.

Also in Table 4, we detected great heterogeneity in the use of the word sex and gender, which were employed as synonyms. In none of these studies is an additional analysis undertaken to try to define gender on any scale and to better analyze the gene-diet-gender interactions at this level. This aspect will, therefore, have to be addressed in newly designed studies. Furthermore, one has to bear in mind that sex and gender also interact. They do so specifically for each health problem analyzed and for each exposure, so that one of the crucial points will be to improve information on the exposure variables and their covariates in nutritional genomics.

## 7. Improvement in Diet Measurement and Other Related Variables Using Gender Perspective in Nutritional Genomics

In nutritional genomics studies one of the major limitations is measuring diet, given that genetic analyses are based on objective laboratory methods, whereas the diet (apart from in strictly controlled intervention studies) is measured through food frequency questionnaires with a strong subjective component that is also subject to recall bias, etc. [105]. Besides that, one has to take into account that most questionnaires have been jointly designed and validated for men and women and are not sex-specific, so that if we take into account that portion size can be different for men and women, as well as other sex-related differences in food consumption [106], new studies would have to be conducted in order to design and validate more sex-specific diet questionnaires as well as to incorporate new technologies in order to improve diet measuring [107]. In parallel, we have stated earlier on that sex and gender interact [108], so in new nutritional genomics studies we would have to pay more attention to measuring variables more related to reproductive health in women, which have so far not been incorporated in studies that do not specifically investigate it. These variables can be critical in understanding the new environmental modulators in the sex-gender interrelationship. For example, information would have to be available on menarche, menopause, full term pregnancy, age of pregnancy, increase in body-weight, whether weight gain changed during pregnancy, undertaking breastfeeding, maternal leave, and care of children, etc. [109,110,111]. There would also have to be more detailed information available on other lifestyle dimensions in which gender differences are important. All of that would provide us with greater knowledge to be incorporated into a better analysis of the gene-diet interactions in nutritional genomics studies with a gender perspective.

## 8. Recommendations on the Design, Analysis, and Presentation of Results in Nutritional Genomics Studies with a Gender Perspective

Following the above-detailed analysis assessing the present situation of studies published on nutritional genomics, we will now make several recommendations, as guidelines, on improving the design, analysis, and presentation of results in nutritional genomics:

a. Get to know the generalities of the gender perspective, the basic concepts and the importance of generating knowledge that will allow us to understand better whether the risk factors of disease (genetic and environmental), their evolution, prognosis and treatments are equal or differ depending on sex/gender.

b. Conduct a literature review of the topic of interest to be investigated including the gender perspective in order to detect the main results as regards genetic and nutritional factors that have an influence on the outcomes and identify the differences per sex/gender that there may be or the gaps that have still not been investigated at this level.

c. Identify whether differences per sex/gender or intermediate phenotypes have been reported in the outcomes and plan the analysis in men and women in accordance with the differences reported by calculating the sample size of each of the groups.

d. Identify whether differences have previously been reported on the effect of the gene variants of interest in the study depending on sex to be able to better design and set the aims of the study and to establish the sample size following those differences. Focus on sex-specific GRS and mechanistic gene-GRS diet interactions.

e. Identify whether differences in the effect of the diet variables of interest in the study per sex/gender have previously been described in order to also take those differences into account in the design of the study, sample size estimation, and statistical analyses.

f. To set the objectives of the study and chose the epidemiological design that will provide the highest level of evidence for the specific characteristics of each investigation and the means available for carrying it out. The study objectives have to consider, mainly as the primary aim, detecting sex/gender differences in the topic that is being analyzed. Ensure adequate representation of males and females and justify reasons for the exclusion of males or females. If it is only possible to include one of the sexes due to the specificity of the health problem analyzed or for funding reasons, this point should be indicated.

g. Choose the methodology for determining sex and gender. Given that nutritional genomics studies often have genome-wide genotyping data available, this data can be used to analyze sex genotypically and compare it with self-reported sex. Although determining gender is more complex, nutritional genomic researchers should work on creating and validating new metrics for measuring gender in these studies.

h. Improve the measuring of diet variables by using the gender perspective (improving food frequency questionnaires specific to sex and introducing new technologies to capture food consumption, etc.).

i. Improve the measuring of other variables related to lifestyle by incorporating the gender perspective to analyze information that has so far not been fully analyzed (reproduction variables in women, etc.)

j. Undertake an analysis plan specifying the statistical analyses to be carried out according to a series of “a priori” and not “a posteriori” hypotheses, testing the gene-diet-sex interactions and undertake and interpret the stratified analyses in accordance with them. The influence of sex/gender should be assessed “a priori” on the basis of their hypothesized role in the gene-diet interactions, causation or impact on the outcome of the health problem analyzed. Doing so minimizes false positives, but one also has to be careful with the false negatives that will have to be interpreted correctly within the statistical power framework of the study.

k. In addition, data should be reported disaggregated by sex/gender (as main or supplemental material, depending on the study aims) regardless of the positive or negative results for future pooling in the meta-analyses.

l. Try to have replication cohorts available in order to analyze the sex-specific results and interactions found to increase the level of evidence if initial results are replicated in other cohorts.

m. Integrate the analyses of other omics to understand the mechanisms better and provide more evidence on the differential effects per sex.

n. Present the results for publication following the guidelines for studies on sex, gender equity in research, such as those of SAGER.

## 9. Conclusions

The incorporation of the gender perspective into nutritional genomics studies will contribute to improving the methodology of those studies as well as the validity of results, and will allow us to obtain new knowledge on gene-diet interactions and other associations that may differ in men and women, thus contributing to making more personalized prevention or treatment possible within the new precision nutrition/precision medicine framework.

## Figures and Tables

**Figure 1 nutrients-11-00004-f001:**
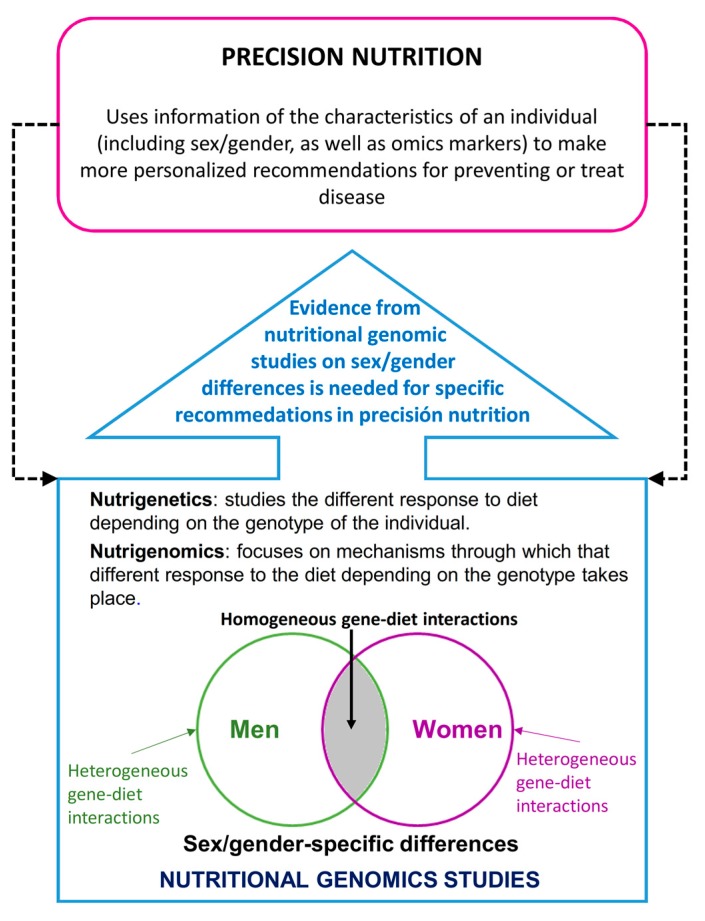
Importance of sex/gender differences in the relationship between nutritional genomics and precision nutrition.

**Table 1 nutrients-11-00004-t001:** Web links containing information related to gender perspective.

Webpage URL
Gendered innovations: http://ec.europa.eu/research/swafs/gendered-innova-tions/index_en.cfm
Gender innovations Stanford University https://genderedinnovations.stanford.edu
eGender platform (Germany): http://egender.charite.de
http://www.euro.who.int/en/health-topics/health-determinants/gender/gender-definitions
The Sex and Gender Medical Education Summit (USA): http://www.sgbmedu-cationsummit.org
The Sex and Gender Women’s Health Collaborative (USA): www.sgwhc.org “Every cell has a sex, and all bodies are influenced by gender”.
The Gender Awakening Tool (Canada): http://www.cwhn.ca/en/node/43342—Sex and Gender in Systematic Reviews: Planning Tool (USA): http://methods.cochrane.org/equity/sex-and-gender-analysis
Toolkit Gender in EU Funded Research (EU): https://publications.europa.eu/es/publication-detail/-/publication/c17a4eba-49ab-40f1-bb7b-bb-6faaf8dec8
The Center for Gender Medicine (CfGM) in the Karolinska Institutet (Sweden): http://ki.se/en/research/centre-for-gender-medicine
Institute of Gender and Health (IGH) of Canadian Institutes of Health Research (Canada): http://www.cihr-irsc.gc.ca/e/48641.html
https://www.who.int/gender-equity-rights/understanding/gender-definition/en/
http://www.gendermedicine.org/index.php?q=node/375 European Society of Gender Health Medicine
http://www.whrn.ca/better-science. Primer published by the Women’s Health Research Network (WHRN), Canada promoting sex and gender-based analyses in health research.
https://www.ag.gov.au/Publications/Documents/AustralianGovernmentGuidelineson theRecognitionofSexandGender/AustralianGovernmentGuidelinesontheRecognition ofSexandGender.pd

**Table 2 nutrients-11-00004-t002:** Sex and gender definitions.

Sex Definition	Gender Definition	Reference
The different biological and physiological characteristics of males and females, such as reproductive organs, chromosomes, hormones, etc.	Refers to the socially constructed characteristics of women and men–such as norms, roles and relationships of and between groups of women and men. It varies from society to society and can be changed—including how they should interact with others of the same or opposite sex within households, communities and work places.	WHO [32]
Refers to a set of biological attributes in humans and animals. It is primarily associated with physical and physiological features including chromosomes, gene expression, hormone levels and function, and reproductive/sexual anatomy. Sex is usually categorized as female or male but there is variation in the biological attributes that comprise sex and how those attributes are expressed.	Refers to the socially constructed roles, behaviours, expressions and identities of girls, women, boys, men, and gender diverse people. It influences how people perceive themselves and each other, how they act and interact, and the distribution of power and resources in society. Gender is usually conceptualized as a binary (girl/woman and boy/man) yet there is considerable diversity in how individuals and groups understand, experience, and express it.	CIHR [33]
Refers to biological differences between females and males, including chromosomes, sex organs, and endogenous hormonal profiles.	Refers to socially constructed and enacted roles and behaviors which occur in a historical and cultural context and vary across societies and over time. All individuals act in many ways that fulfill the gender expectations of their society. With continuous interaction between sex and gender, health is determined by both biology and the expression of gender.	NIH [34]
Biological and physiological characteristics that define humans as female or male.	Social attributes and opportunities associated with being female and male and to the relationships between women and men and girls and boys, as well as to the relations between women and those between men.	EIGE [35]
Refers to the chromosomal, gonadal and anatomical characteristics associated with biological sex.	It is a part of a person’s personal and social identity. It refers to the way a person feels, presents and is recognized within the community. A person’s gender may be reflected in outward social markers, including their name, outward appearance, mannerisms and dress.	Australian Government [36]

WHO: World Health Organization; CIHR: Canadian Institutes of Health Research; NIH: National Institutes of Health; EIGE: European Institute of Gender Equality.

**Table 3 nutrients-11-00004-t003:** Examples of Sex-specific SNPs associated with different traits.

Author	SNP/Loci/Gene	Trait	Sex-Specific	Chromosome
Tukiainen et al., 2014 [63]	ITM2A, FGF16 ATRX/MAGT1 (Xq23) (no associated gene)	Height Fasting insulin levels	Female (height) Male (fasting insulin)	X
Russo Paola et al., 2008 [64]	TBL1Y, USP9Y	Lower triglycerides (TG) and higher HDL	Male	Y
Estrada et al., 2012 [65]	Rs5934507 (Xp22.31, closest gene FAM9B/KAL1	Bone metabolism	Male	X
Ohlsson et al., 2011 [66]	rs5934505 in the X Chromosome near FAM9B	Lower testosterone levels	Male	X
Winkler et al., 2015 [67]	44 loci	Changes in body mass index (BMI), measures of body size, or waist-to-hip ratio	Female (28) and male (5) and 11 opposite effects	Autosome chromosomes
Randall et al., 2013 [68]	GRB14/COBLL1, LYPLAL1/SLC30A10, VEGFA, ADAMTS9, MAP3K1, HSD17B4, PPARG	Anthropometrics (specifically waist phenotypes)	Female	2,1,6,3,5,5,3
Heid et al., 2010 [69]	RSPO3, VEGFA, GRB14, LYPLA1, ITPR2-SSPN, ADAMTS9	Waist-to-hip ratio	Female	6,6,2,1,12,3
Porcu et al., 2013 [70]	PDE8B, PDE10A, MAF/LOC440389, LPCT2/CAPNS2, NETO1/FBXO15	Affect thyroid stimulating hormone (TSH) and Free thyroxine (FT4) levels	Male (first 4) and Female (last 1)	5,6,16,16,18
Kitamoto et al., 2015 [71]	ADIPOQ (2 SNPs out of 7)	Decrease serum adiponectin levels which are correlated with decreased homeostatic model assessment-insulin resistance (HOMA-IR)	Female	3
Mittelstrass et al., 2011 [72]	CSP1	Sex-specific association with glycine	Female and male	2
Teslovich et al., 2010 [73]	KLF14, ABCA8	Sex-specific association with TG, LDL-C	Female	7,17
Aung et al., 2014 [74]	ZNF259	Many lipid associations	Male and Female	11
Lee, Kwon, & Park, 2017 [75]	CYP11β2	Salt sensitive gene leads to obesity from salt-intake?	Female	8
Nilsson et al., 2011 [76]	PRL	Sex-specific obesity	Male	6
Sun et al., 2014 [77]	GCKR, SLC2A9, SF1	Associated with serum uric acid concentrations	Male	2,4,9
Sung et al., 2016 [78]	BBS9, ADCY8, KCNK9 MLLT10/DNAJC1/EBLN1	Visceral fat Subcutaneous fat	Women	10,8

**Table 4 nutrients-11-00004-t004:** Selected studies analyzing sex-specific gene-diet interactions in determining different phenotypes.

Author	Study Characteristics and Aims	Findings
Ordovas JM et al., 2002 [85]	Examined whether dietary fat modulates the association between the APOA1 G-A polymorphism and HDL-in men and women from the Framingham Study.	We found a significant gene-diet interaction associated with the APOA1 G-A polymorphism. In women carriers of the A allele, higher polyunsaturated fatty acid (PUFA) intakes were associated with higher HDL-cholesterol concentrations, whereas the opposite effect was observed in G/G women.
Ribalta J et al., 2005 [86]	Identification of gender-specific genetic influences on fasting and postprandial TG concentrations under typical living conditions in healthy, lean, normolipidemics.	An adverse combination of common alleles of the FABP-2, APOE, and PPARgamma genes in women increases their TGs to values comparable to those seen in men. Although this influence is not appreciable when studying fasting plasma TGs, it becomes apparent with use of a more sensitive index.
Méplan C et al., 2007 [87]	Analysis of polymorphisms in the selenoprotein P gene determine the response of selenoprotein markers to selenium (Se) supplementation in a gender-specific manner (the SELGEN study).	Two common functional SNPs within the human SePP gene that may predict behavior of biomarkers of Se status and response to supplementation and thus susceptibility to disease. Both SNPs and gender were associated with differences in scavenger glutathione peroxidase 3 (GPx3) activity and other markers.
Hu M et al., 2012 [88]	Intervention with a high-carbohydrate (high-CHO) diet for a short-term and investigation of the interactions with the hepatic lipase G-250A promoter polymorphism to affect the ratios of plasma lipids and apolipoproteins.	The high-CHO diet induced the positive effects on the lipid ratios in general, only except the TG/HDL-C ratio in females. Noticeably, the decreased apoB100/apoAI ratio was associated with the A allele of hepatic lipase G-250A polymorphism only in males.
Shatwan IM et al., 2016 [89]	Analysis of the influence of two commonly studied LPL polymorphisms (rs320, HindIII; rs328, S447X) on postprandial lipaemia, in 261 participants using a standard sequential meal challenge.	Novel finding on the effect of the LPL S447X polymorphism on the postprandial glucose and gender-specific impact of the polymorphism on fasting and postprandial TAG concentrations in response to sequential meal challenge in healthy participants. The sex-specific results were only detected in men.
Jacobo-Albavera L et al., 2015 [90]	Analysis of whether gender, menopausal status and macronutrient proportions of diet modulate the effect of the (ABCA1) R230C variant on various metabolic parameters.	First study reporting a gender-specific interaction between ABCA1/R230C variant and dietary carbohydrate and fat percentages affecting Visceral adipose tissue (VAT) / subcutaneous adipose tissue (SAT) ratio, gamma-glutamyl transpeptidase (GGT), alkaline phosphatase (ALP), adiponectin levels and HOMA index. This study confirmed a previously reported gender-specific ABCA1-diet interaction affecting HDL-C levels.
Zhang Z et al., 2011 [91]	Investigation of the association between the sterol regulatory element-binding protein-1c gene (SREBP-1c) rs2297508 and the changes in lipid profiles in a high-carbohydrate and low-fat diet in a Chinese population.	The C allele of the rs2297508 polymorphism was associated with a retardation of the increases in serum triacylglycerol, serum insulin, and HOMA-IR in females and with the elevated serum HDL-C in males after the high-carbohydrate/low fat (high-CHO/LF) diet.
Barragán R et al., 2018 [92]	Analyzed the age influence on the intensity rating of the five basic tastes: sweet, salty, bitter, sour and umami (separately and jointly in a "total taste score") and their modulation by sex and genetics in a relatively healthy population.	Women perceived taste significantly more intense than men (*p* = 1.4 × 10^−8^ for total taste score). Significant associations were, found between a higher perception of sour taste and a higher preference for it in women. In contrast, the higher perception of sweet was significantly associated with a higher preference for bitter in both, men and women. The TAS2R38-rs713598 SNP had a significant interaction with sex.
Lauritzen L et al., 2017 [93]	Mendelian randomization study to explore whether SNPs in fatty acid desaturase (FADS) and elongase (ELOVL) genes were associated with school performance in a sex-specific manner.	Associations between rs1535 minor allele homozygosity and rs174448 major allele carriage and improved performance in boys but not in girls was found, thereby counteracting existing sex differences.
Obregón AM et al., 2017 [94]	Analysis of the association between the DRD2 rs1800497 polymorphism and eating behavior in Chilean children.	In the sex-specific analysis, the TaqI A1 allele was associated with higher scores on Satiety Responsiveness and Emotional Undereating subscales in obese girls, and higher scores of Enjoyment of Food subscale in boys.
Roumans NJ et al., 2015 [95]	Investigation of whether genetic variation in extracellular matrix (ECM) -related genes is associated with weight regain among participants of the European DiOGenes study.	Variants of ECM genes were associated with weight regain after weight loss in a sex-specific manner.
Ericson U et al., 2013 [96]	Interaction analysis between IRS1 rs2943641 and macronutrient intakes on incident T2D and percentage body fat in the Malmö Diet and Cancer cohort.	The IRS1 rs2943641 interacted with carbohydrate and fat intakes on incident T2D in a sex-specific manner. A protective association between the rs2943641 T allele and T2D was restricted to women with low carbohydrate intake and to men with low fat intake.
Dedoussis GV et al., 2011 [97]	Investigation of the age-related association between the Pro12Ala variant (rs1801282) and diet in obesity-related traits in children.	Adiposity in children was influenced by the Pro12Ala polymorphism in a sex-specific and age-dependent manner.
Nettleton JA et al., 2009 [98]	Analysis of whether dietary macronutrient intake modified associations between ANGPTL4[E40K] variation and TG and HDL-C in White men and women from the Atherosclerosis Risk in Communities study.	In men, but not women, the inverse association between carbohydrate and HDL-C was stronger in A allele carriers (beta+/-S.E. -1.80+/-0.54) than non-carriers (beta+/-S.E. -0.54+/-0.11, p(interaction) = 0.04 in men and 0.69 in women; p 3-way interaction = 0.14).
Gastaldi M et al., 2007 [99]	Analysis of the effect of fatty acid binding protein 2 (FABP2) Ala54Thr and microsomal triacylglycerol transfer protein (MTTP) -493G/T variations on plasma lipid markers, at baseline and on the response to the 3-mo Medi-RIVAGE study.	These 2 polymorphic loci are thus differently associated with the baseline lipid markers as well as with the response to nutritional recommendations, but both presented a marked sex-specific profile, with the response to diet being particularly efficient in men homozygous for the MTTP -493T allele.
Alkhalaf A et al., 2015 [100]	This study investigated whether 5L-5L in the CNDP1 gene was associated with mortality and progression of renal function loss and to what extent this effect was modified by sex.	The association between CNDP1 and cardiovascular mortality was sex-specific, with a higher risk in women with 5L-5L genotype. CNDP1 was not associated with all-cause mortality or change in epidermal growth factor receptor (EGFR).
van Dijk SJ et al., 2016 [101]	A double-blind randomized placebo-controlled trial in pregnant women to test whether a defined nutritional exposure in utero, docosahexaenoic acid (DHA), could alter the infant epigenome.	Maternal DHA supplementation during the second half of pregnancy had small effects on DNA methylation of infants. However, the number of differential methylated loci (DMRs) at birth was greater in males (127 DMRs) than in females (72 DMRs) separately, indicating a gender-specific effect.
Gonzalez-Nahm S et al., 2017 [102]	Association between maternal adherence to a Mediterranean diet pattern during pregnancy and infant DNA methylation at birth.	There was an association between overall diet pattern and methylation at the 9 DMRs analyzed and suggests that maternal diet can have a sex-specific impact on infant DNA methylation at specific imprinted DMRs.
Borgo F et al., 2018 [103]	Comparison of the gut microbiota is in at least two separate microbial populations, the lumen-associated (LAM) and the mucosa-associated microbiota (MAM). Next generation sequencing was used).	LAM and MAM communities seemed to be influenced by different host factors, such as diet and sex. Female MAM was enriched in Actinobacteria (with an increased trend of the genus *Bifidobacterium*), and a significant depletion in *Veillonellaceae*.
Bolnick DI et al., 2014 [104]	Analysis of the factors related to the composition of gut microbiota, mainly the host genotyping.	The results indicated that microbiota composition depends on interactions between host diet and sex within populations of wild and laboratory fish, laboratory mice and humans. The diet-microbiota associations were sex dependent. Further experimental work confirmed that microbiota was different males versus females and suggested that therapies to treat dysbiosis might have sex-specific effects

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
