# Peer review of "A Guide to Applying the Sex-Gender Perspective to Nutritional Genomics"

_nutrients, 2018, doi:10.3390/nu11010004_

Reviewer 1 Report

The purpose of precision nutrition is to make dietary recommendations of a more personalized nature possible so as to optimize the prevention or delay of a disease and to improve health. Authors analyze the main sex-specific gene-diet interactions published to date and their main limitations and present guidelines with recommendations to be followed in new nutritional genomics studies incorporating the gender perspective. This manuscript has written well, however, minor spell chech will need before publication.

Author Response

[R1] The purpose of precision nutrition is to make dietary recommendations of a more personalized nature possible so as to optimize the prevention or delay of a disease and to improve health. Authors analyze the main sex-specific gene-diet interactions published to date and their main limitations and present guidelines with recommendations to be followed in new nutritional genomics studies incorporating the gender perspective. This manuscript has written well, however, minor spell chech will need before publication.

 We would like to thank reviewer 1 for his/her general comments on our article. We have re-checked the article with the help of a native English speaker, an expert on this theme, and corrected the mistakes mentioned.

Reviewer 2 Report

Well written, interesting article, employing appropriate methodology, that will contribute to the literature.

Author Response

[R2] Well written, interesting article, employing appropriate methodology, that will contribute to the literature.

Thanks to reviewer 2 for his/her comments on our article. We have re-checked the entire article with the help of a native English speaker and have corrected the mistakes mentioned.

Reviewer 3 Report

This was a thorough review and argument of why we (scientists) need to focus more on genomics and gender. This was a lengthy review and the reviewer just have a few suggestions to enhance the first portion of this manuscript:

Abstract: The first 3 lines read similar to an introduction. Suggest including a line about nutritional genomics, the lack of considering sex and then the methodology, etc.

 Introduction:

Part 1, There is a lot of information in this portion to explain personalized nutrition and the argument for including gender. Since genomics is complex, would suggest there is a figure to visually aid the reader in understanding why personalized nutrition and why sex is necessary to include. The reviewer suggests this part either be broken down into further sub-sections to discuss all aspects of genomics and the link with personalized nutrition or else focus on just 1 aspect.

 Part 2, the one argument that is overlooked in this paragraph is that for clinical trials, people make that decision on whether or not they want to participate in studies. Thus, in an ideal world, it would be great to have a mix of males and females, but realistically this may not occur. Also, if researchers focus on breast cancer, more females may be diagnosed with this than males. This goes the same for other diseases as well. Thus, if a balance of males and females were to be studied, the population size may be small, which may also limit the generalizability.

 Part 3, the paragraph about the difference between gender and sex was quite long. This could have been embedded within part 2 and thus eliminating part 3 altogether. This would then cause the nutritional genomic information to be placed in part 2.

Lines 231-251: may consider reducing the information here.

Author Response

[R2] This was a thorough review and argument of why we (scientists) need to focus more on genomics and gender. This was a lengthy review and the reviewer just have a few suggestions to enhance the first portion of this manuscript:

First of all, we would like to thank reviewer 3 for his/her general comments on our article and also for his/her suggestions on enhancing the first portion of this manuscript. We have taken all his/her comments into consideration and provide details on all the changes made in our responses to each specific suggestion. We have marked all these changes to the manuscript in red. The manuscript has now been checked by a native English speaker, an expert on this theme. All the changes necessary to overcome the limitations pointed out by the reviewer have been made.

[R2] Abstract: The first 3 lines read similar to an introduction. Suggest including a line about nutritional genomics, the lack of considering sex and then the methodology, etc.

We have revised the abstract and included a line about nutritional genomics as the reviewer suggested.

[R2] Introduction:

Part 1, There is a lot of information in this portion to explain personalized nutrition and the argument for including gender. Since genomics is complex, would suggest there is a figure to visually aid the reader in understanding why personalized nutrition and why sex is necessary to include. The reviewer suggests this part either be broken down into further sub-sections to discuss all aspects of genomics and the link with personalized nutrition or else focus on just 1 aspect.

In line with the reviewer’s suggestion, we have added a Figure (Figure 1) so as to help the reader understand the concepts. We have also broken part 1 down into further subsections (1.1 and 1.2) so as to make it clearer. See lines 48 to 105.

[R2] Part 2, the one argument that is overlooked in this paragraph is that for clinical trials, people make that decision on whether or not they want to participate in studies. Thus, in an ideal world, it would be great to have a mix of males and females, but realistically this may not occur. Also, if researchers focus on breast cancer, more females may be diagnosed with this than males. This goes the same for other diseases as well. Thus, if a balance of males and females were to be studied, the population size may be small, which may also limit the generalizability.

We are grateful to the reviewer for this suggestion. In the new version of the manuscript we have included a new paragraph in part 2 that deals with the argument overlooked in the original version. See lines 194 to 200.

[R2] Part 3, the paragraph about the difference between gender and sex was quite long. This could have been embedded within part 2 and thus eliminating part 3 altogether. This would then cause the nutritional genomic information to be placed in part 2.

Although we understand the reviewer’s concern, we believe that part 2 is already very long and complex. If we eliminate part 3 and add it to part 2, this would only make part 2 even longer and more complex. We think, therefore, that it would be better to leave the structure as it is. We have, nevertheless checked the paragraph that the reviewer refer to, and despite being long, it is a very important point that has to be clearly explained. 

[R2] Lines 231-251: may consider reducing the information here.

We have checked the lines of the text indicated by the reviewer and shortened them in the new version of the manuscript, as the reviewer suggests. See lines 244 to 261.
